# Collapse of layer dimerization in the photo-induced hidden state of 1T-TaS$_2$

Quirin Stahl[1], Maximilian Kusch[1], Florian Heinsch[1,2], Gaston Garbarino [3], Norman Kretzschmar[3],
Kerstin Hanff[4], Kai Rossnagel [4,5,6], Jochen Geck[1,7] & Tobias Ritschel [1✉]

Photo-induced switching between collective quantum states of matter is a fascinating rising field with exciting opportunities for novel technologies. Presently, very intensively studied examples in this regard are nanometer-thick single crystals of the layered material 1T-TaS$_2$, where picosecond laser pulses can trigger a fully reversible insulator-to-metal transition (IMT). This IMT is believed to be connected to the switching between metastable collective quantum states, but the microscopic nature of this so-called hidden quantum state remained largely elusive up to now. Here, we characterize the hidden quantum state of 1T-TaS$_2$ by means of state-of-the-art x-ray diffraction and show that the laser-driven IMT involves a marked rearrangement of the charge and orbital order in the direction perpendicular to the TaS$_2$-layers. More specifically, we identify the collapse of interlayer molecular orbital dimers as a key mechanism for this non-thermal collective transition between two truly long-range ordered electronic crystals.

[1] Institut für Festkörper- und Materialphysik, Technische Universität Dresden, 01069 Dresden, Germany. [2] Institute of Radiation Physics, Helmholtz-Zentrum Dresden-Rossendorf, 01328 Dresden, Germany. [3] ESRF, The European Synchrotron, 38000 Grenoble, France. [4] Institut für Experimentelle und Angewandte Physik, Christian-Albrechts-Universität zu Kiel, 24098 Kiel, Germany. [5] Ruprecht-Haensel-Labor, Christian-Albrechts-Universität zu Kiel und Deutsches Elektronen-Synchrotron DESY, 24098 Kiel und, 22607 Hamburg, Germany. [6] Deutsches Elektronen-Synchrotron DESY, 22607 Hamburg, Germany. [7] Würzburg-Dresden Cluster of Excellence ct.qmat, Technische Universität Dresden, 01069 Dresden, Germany. ✉email: tobias.ritschel@tu-dresden.de

The layered transition metal dichalcogenides (TMDs) form a vast class of materials hosting diverse non-trivial quantum phenomena such as spin-valley polarization[1], Ising-superconductivity[2], or intertwined electronic orders[3,4]. All these intriguing electronic effects along with the natural suitability of TMDs for the preparation of quasi two-dimensional (2D) nanosheets render them highly appealing for next-generation technologies[5–8].

1T-TaS$_2$ is a particularly interesting and extensively studied TMD in which external tuning parameters like temperature, pressure or chemical substitution span a very complex electronic phase diagram. Apart from several charge density waves (CDWs) this phase diagram also features pressure-induced super-conductivity and a so-called Mott-phase, which stands out due to its semiconducting electronic transport properties[3,9].

Remarkably, besides the aforementioned states that can be reached in thermal equilibrium, femto to picosecond optical or electrical pulses can trigger a non-equilibrium IMT into a previously hidden and persistent metallic CDW-state[7,10,11]. The discovery of this so-called hidden CDW (HCDW) has sparked wide excitement as it might provide a new platform for memory device applications. Accordingly, in recent years, a significant number of experimental and theoretical studies aimed at pinning down the microscopic mechanism of this non-equilibrium IMT that is believed to be connected to a reorganization of the CDW-order.

Although significant efforts to determine the microscopic processes underlying this novel IMT have been made[12–17], a clear picture remains elusive.

In this article we address this open issue directly by means of high-resolution synchrotron x-ray diffraction (XRD) in combination with laser pumping. Our experiments enable the examination of the laser-driven transition and in particular of the HCDW-order in 1T-TaS$_2$ nanosheets with great sensitivity and in all three spatial directions. In this way we show that ultra-short laser pulses drive a collapse of the interlayer dimerization present in thermal equilibrium, which reveals the key physical process behind the laser-driven IMT of 1T-TaS$_2$ .

## Results

**CDW-states in thermal equilibrium.** At ambient pressure and at temperatures below ≈180 K a commensurate charge density wave (CCDW) develops within the layers of 1T-TaS$_2$, which is characterized by the formation of star-of-David (SOD) shaped clusters containing 13 Ta-sites arranged in a $\sqrt{13} \times \sqrt{13}$ superlattice (cf. Fig. 1a, b). The periodic lattice distortion associated with a CDW has a larger periodicity than the underlying crystal lattice and becomes visible in XRD through the appearance of characteristic superlattice reflections which are typically $\geq 10^2$ times weaker than the Bragg reflections stemming from the average

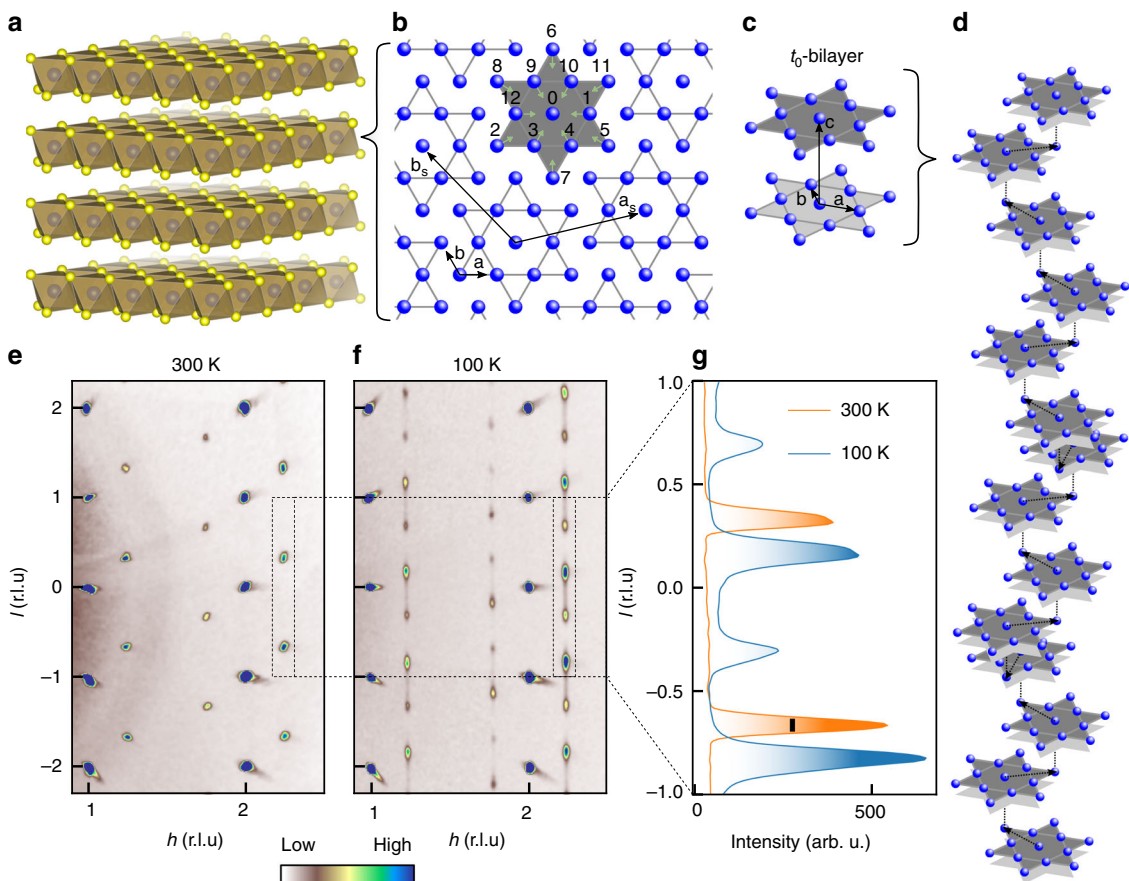

**Fig. 1 Structure and CDW layer stacking in 1T-TaS$_2$. a** The layered host structure of 1T-TaS$_2$ comprises S-Ta-S layers (Spacegroup: $P\overline{3}m1$). **b** The structural key feature of the C and NCCDW are star-of-David (SOD) shaped clusters containing 13 Tantalum sites formed by an inward displacement (green arrows) of 12 Ta ions (labels 1…12) towards the central Ta ion (label 0). The SOD clusters themselves form a $\sqrt{13} \times \sqrt{13}$-superlattice with in-plane lattice vectors $a_s$ and $b_s$. **c, d** The partially disordered stacking of the CCDW at low temperatures is dominated by $t_0$ bilayers (**c**) which by themselves are stacked with a vector randomly drawn from the three symmetry equivalent vectors $t_7$, $t_8$ and $t_{11}$ (**d**). The $t_0$ bilayers are illustrated as gray SODs in **d**. **e, f** Reciprocal space maps parallel to the $h0l$-plane (integration thickness perpendicular to the plane: $\delta k = 0.3$ r.l.u) for T = 300 K (**e** NCCDW) and T = 100 K (**f** CCDW). **g** Intensity profile of the superlattice reflections along the $l$-direction for CCDW (blue) and NCCDW (orange). The double-peak feature is the key fingerprint for the $t_0$ bilayers present in the CCDW. The black bar indicates the full width at half maximum of a typical Bragg peak.

structure. For the CCDW these reflections occur at the commensurate $q$-vectors $q_1^C = (\sigma_1^C, \sigma_2^C, l)$ and $q_2^C = (-\sigma_2^C, \sigma_1^C + \sigma_2^C, l)$ with $\sigma_1^C = 3/13$ and $\sigma_2^C = 1/13$[18,19]. The superlattice structure of the CCDW is very well ordered within the *ab*-plane which translates into sharp superlattice reflections in the *hk*-plane. However, in the direction perpendicular to the S-Ta-S-layers, the CDW order is subject to partial disorder among the 13 possible two-layer stacking arrangements labeled as $t_0...t_{12}$ corresponding to the 13 Ta-sites within one SOD cluster (cf. Fig. 1b).

Due to this disorder in one direction the superlattice reflections form diffuse stripes along the *l*-direction rather than sharp peaks. The intensity distribution along these stripes has a characteristic two-peak structure (see Fig. 1f, g). A careful analysis of the diffuse-scattering reveals that the 3D arrangement of the CDW in 1T-TaS$_2$ at low temperatures is given by a stacking in which bilayers obeying the $t_0$-stacking occur, as illustrated in Fig. 1c. These bilayers are arranged in a stacking sequence, where the stacking vector between two adjacent bilayers is randomly chosen from the three symmetry equivalent vectors $t_7$, $t_8$, and $t_{11}$, which is schematically depicted in Fig. 1d[20–23].

It should be noted that merely the superlattice reflections show diffuse-scattering stripes while the Bragg-peaks remain sharp, indicating the absence of more conventional stacking faults in the average structure.

The CCDW phase is also often referred to as the Mott-phase, because its unusual semiconducting transport properties are widely believed to stem from Mott-Hubbard type electron-electron correlations[24]. But this paradigm is currently under intense debate as several studies find that the CDW-stacking in the -out-of-plane direction has a marked influence on the low-energy electronic structure. In fact, the CDW-stacking can also drive the formation of a charge excitation gap at the Fermi energy and, hence, yield the observed semiconducting transport behavior. The crucial ingredient for the stacking-induced gap-formation seems to be the occurrence of the $t_0$-bilayers[4,23,25,26].

Increasing the temperature or applying external pressure transforms the CCDW into a so-called nearly commensurate CDW (NCCDW), accompanied by a significant metallization of the sample. The NCCDW is structurally characterized by the formation of a well-ordered network of defects—so-called discommensurations—within the $\sqrt{13} \times \sqrt{13}$-superlattice of SODs[19,27,28]. This in-plane modification of the CDW structure takes place on a length-scale of about 70 Å and is accompanied by a marked rearrangement of the CDW order in the out-of-plane direction. In XRD, the fingerprints of the NCCDW are given by a slight shift of the $q$-vectors to $q_1^{NC} = (\sigma_1^{NC}, \sigma_2^{NC}, 1/3)$ and $q_2^{NC} = (-\sigma_2^{NC}, \sigma_1^{NC} + \sigma_2^{NC}, -1/3)$ with $\sigma_1^{NC} = 0.248$ and $\sigma_2^{NC} = 0.068$[19], along with the appearance of strong higher-order superlattice reflections owing to the domain-like structure imposed by the discommensurations. Instead of the diffuse stripes along the *l*-direction which occur for the CCDW, the superlattice reflections of the NCCDW are sharp and centered at $l_{NC} = \pm 1/3$, indicating a well-ordered CDW stacking with a periodicity of three times the interlayer distance and the absence of the $t_0$ bilayers (cf. Fig. 1e, g). Although several scenarios for the microscopic mechanism of the IMT associated with the NCCDW-CCDW transition are currently being discussed, there is increasing theoretical and experimental evidence that exactly this rearrangement of the CDW stacking plays a crucial role[23,26,29].

Since the photoinduced HCDW is also accompanied by a metallization of 1T-TaS$_2$, the question arises whether or not the CDW stacking in the out-of-plane direction is affected by this transition, too. In order to address exactly this issue, we performed XRD on thin exfoliated samples of 1T-TaS$_2$ in which we induced the HCDW by means of a single 1.6 ps pulse from a Ti-sapphire laser. The results of these measurements will be presented in the following section.

**Photo-induced metastable CDW-state.** The presented experiments have been performed on thin flakes exfoliated from high-quality 1T-TaS$_2$ single crystals at the beamline ID09 at the ESRF (see "methods" for details).

In Fig. 2, we summarize the key results of these measurements by showing reconstructed reciprocal space maps for a switching cycle

$$(\text{NCCDW}) \overset{\text{cooling}}{\rightarrow} (\text{CCDW}) \overset{\text{laser pulse}}{\rightarrow} (\text{HCDW})$$

of a thin flake 1T-TaS$_2$ sample. Figure 2a–c illustrate the changes of the in-plane components of the superlattice reflections by projecting the diffraction pattern along the *l*-direction within a range of $-1/3 < l < 1/3$ onto the *hk*0-plane, i. e. the plane parallel to the S-Ta-S layers. In order to visualize the changes of the out-of-plane component and the peak profile of superlattice reflections in the out-of-plane direction, we project the diffraction pattern onto the *h*0*l*-plane within a *k*-range of $-0.1 < k < 0.1$ in Fig. 2e, f.

Starting at room temperature, we observe the characteristic higher-order superlattice reflections in the projection onto the *hk*-plane, which indicates the presence of ordered discommensurations in the NCCDW (see Fig. 2a). The in-plane-parameters of the $q$-vector $\sigma_1^{NC} = 0.245 \pm 0.002$ and $\sigma_2^{NC} = 0.067 \pm 0.002$ are in excellent agreement with the values obtained for single crystal bulk samples[19,30]. Moreover, for these thin flakes the NCCDW superlattice relections are sharp along the *l*-direction and appear at $l = \pm 1/3$ indicating the tripling of the unit cell in the out-of-plane direction (Fig. 2(g)). After cooling the sample to 4 K the higher-order superlattice reflections vanish as the in-plane components of the $q$-vector assume the commensurate values $\sigma_1^C = 3/13$ and $\sigma_2^C = 1/13$. (see Fig. 2b). As can be observed in Fig. 2e, h, the superlattice peaks exhibit characteristic diffuse stripes with a double-peak structure along the *l*-direction. This is also found for bulk single crystals and indicates the peculiar CDW-stacking of the CCDW described above. We can therefore conclude that our thin exfoliated flakes of 1T-TaS$_2$ and bulk single crystals of 1T-TaS$_2$ develop the same in-plane CDW-structure and out-of-plane CDW-stacking.

Having established that the diffraction pattern unambiguously reveals the formation of the CCDW at low temperatures, we continued by photoexcitation of the system. At a nominal temperature of 4 K, we excited the sample with a single 1.6 ps long laser pulse with a wavelength of 800 nm and a pulse energy of 0.18 µJ which corresponds to a fluence in the range of 1 mJ/cm$^2$. According to previous reports, this procedure creates the HCDW[7]. Indeed, in response to the photoexcitation the diffraction pattern changes very clearly and the higher-order superlattice peaks in the projection onto the *hk*-plane appear again. This is qualitatively similar to the NCCDW at room temperature, yet with somewhat different in-plane components of the $q$-vector, namely $\sigma_1^H = 0.243 \pm 0.002$ and $\sigma_2^H = 0.070 \pm 0.003$ and, hence, a smaller discommenurability $\delta = |q_H^\| - q_C^\||$ as compared with the NCCDW (see also Table 1 for a comparison of the in-plane $q$-vector parameters). Even more striking is that the peak profile in the *l*-direction becomes sharp again and shifts back to $l = \pm 1/3$. In other words, the double-peak structure, characteristic for the $t_0$-bilayers in the CCDW, disappears upon laser pumping. This strong effect of the laser pump pulse can be clearly observed in Fig. 2e, f, h, i.

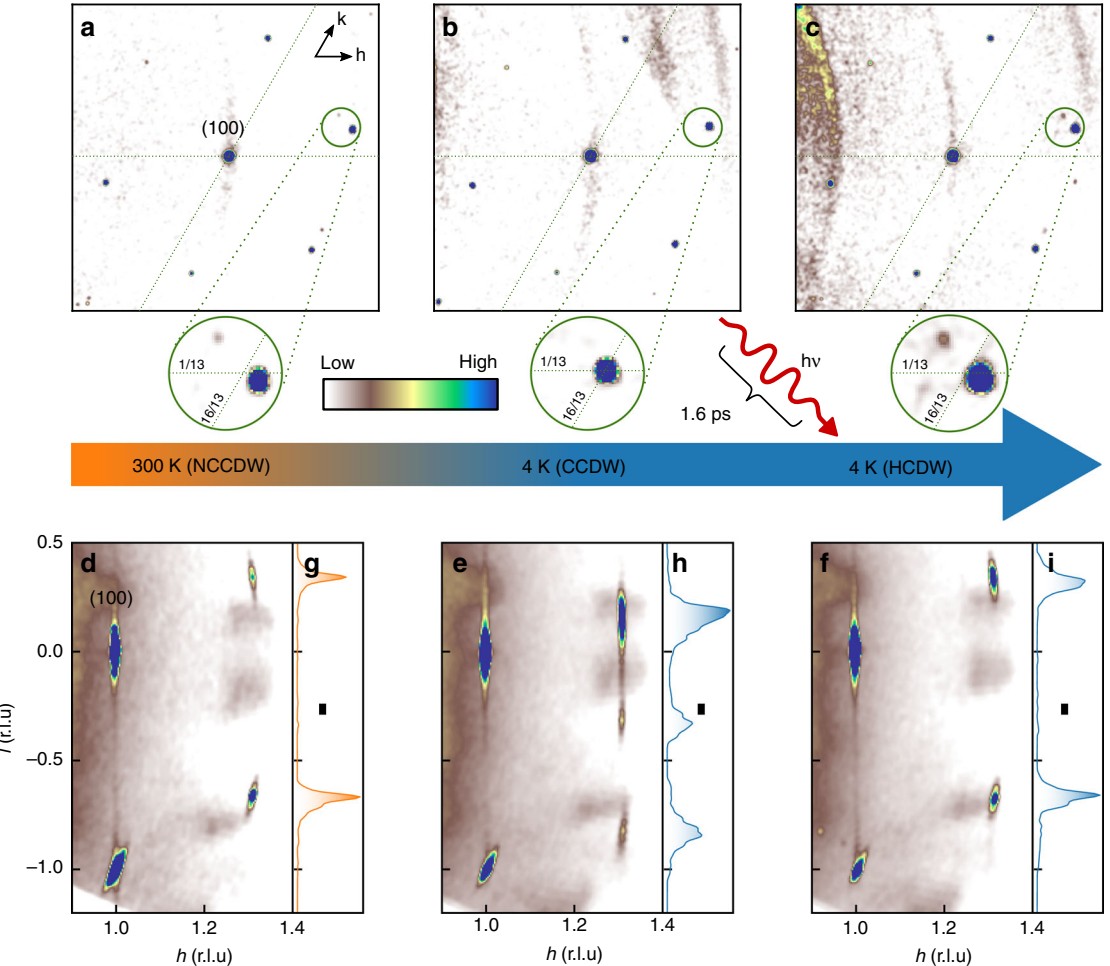

**Fig. 2 Photo-induced changes of the XRD from a thin flake of 1T-TaS$_2$ .** Reciprocal space maps parallel to the *hk*0-plane (integration thickness perpendicular to the plane: $\delta l = 2/3$ r.l.u.) for the NCCDW at room temperature (**a**), the CCDW at $T = 4$ K (**b**) and the photo-induced HCDW at the same temperature $T = 4$ K. The characteristic occurrence of strong higher-order superlattice reflections due to the in-plane discommensurations network is clearly observable for the NCCDW (**a**) and the HCDW (**c**). Reciprocal space maps parallel to the *h*0*l*-plane (integration thickness perpendicular to the plane: $\delta k = 0.3$ r.l.u.) are shown in **d**–**f** for NCCDW, CCDW, and HCDW, respectively. The extracted intensity distribution of the superlattice reflections along the *l*-direction clearly shows the well-defined sharp spots for the NCCDW (**g**), the characteristic diffuse double-peak feature for the CCDW (**h**) and the vanishing of the diffuse scattering for the photo-induced HCDW. The black bars in **g**, **h**, and **i** indicate the full width at half maximum of a typical Bragg peak.

**Table 1 Parameters of the in-plane *q*-vector for CCDW, HCDW, and NCCDW.**

| state | $\sigma_1$ (r.l.u.) | $\sigma_2$ (r.l.u.) | $\delta$ ($10^{-2}$ r.l.u.) | $\phi$ (deg) | $|q|$ (r.l.u.) |
|-------|--------------------|--------------------|----------------------------|-------------|----------------|
| C | 3/13 | 1/13 | 0 | 13.898 | 0.277 |
| H | 0.243 ± 0.002 | 0.070 ± 0.003 | 1.06 ± 0.16 | 12.3 ± .3 | 0.284 ± 0.002 |
| NC | 0.245 ± 0.002 | 0.067 ± 0.002 | 1.32 ± 0.16 | 11.9 ± 0.3 | 0.284 ± 0.002 |

$\sigma_1$ and $\sigma_2$ are the in-plane components. $\delta$ is the incommensurability $\delta = |q_H^{\parallel} - q_C^{\parallel}|$. $\phi$ is the angle of the in-plane *q*-vector with respect to the reciprocal $a^*$ axis and $|q|$ is the length of the in-plane *q*-vector.

For all three states the superlattice peak widths within the plane are always comparable with the width of the Bragg peaks, which indicates that the in-plane superstructure remains long-range ordered. Note that the XRD data for the HCDW has been taken several seconds after the laser pulse hit the sample, which means that the sample temperature is well-defined at $T = 4$ K. The fact that the CCDW would be the thermodynamic stable state under these conditions points to the metastable nature of the HCDW. Furthermore, the resolution limited peak width observed here, shows that the non-equilibrium process driven by the 1.6 ps laser pulse results in a truly long-range ordered HCDW-state.

**Temperature driven relaxation.** In Fig. 3 we show the evolution of the superstructure peaks for two successive laser pulse/ heating cycles on the same sample. As can be clearly seen in Fig. 3e, f, j, k the sharp peak associated with the HCDW disappears in a temperature range around 40 K and the system flips back to the CCDW. This is indeed a very characteristic property of the HCDW which was found in previous resistivity measurements[7]. Notably, this temperature is significantly lower than the thermally driven CCDW-NCCDW transition, at about 200 K. Our observation of the same behavior provides strong evidence that the photo-induced state created in our

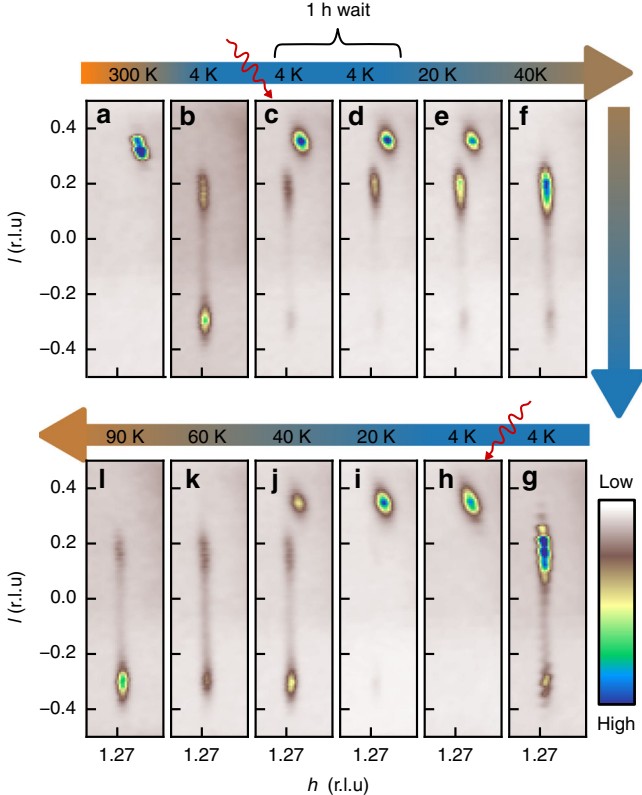

**Fig. 3 Thermally driven HCDW-CCDW transition for two successive laser pulse/heating cycles. a** The NCCDW superstructure peak at room temperature. **b** The diffuse CCDW double peak at 4 K. **c** The peak $l$-profile after the first photoexcitation at 3 K. Coexistence of CCDW peak and HCDW peak indicating only partial switching. **d** Same as **c** after waiting for 1 h, demonstrating the persistence of the HCDW. **e**, **f** Peak $l$-profiles for heating up to 40 K where the HCDW peak vanishes. **g** CCDW peak $l$-profile after cooling to 4 K again. **h** Peak $l$-profile of the HCDW after a second laser pulse with higher fluence resulting in a more complete switching. **i**–**l** Second warming cycle up to 90 K. The CCDW peaks reappear at around 40 K, in **j**.

experiment is indeed the hidden state which had been reported previously.

We also mention that for some samples we observed an incomplete photo-induced switching from the CCDW to the NCCDW, resulting in the simultaneous occurrence of both types of corresponding superlattice reflections (cf. Fig. 3c). This observation, which we attribute to a slight mismatch of sample size, laser spot size and x-ray spot size, demonstrates that both types of long-range orders—the CCDW and the HCDW—can macroscopically coexist in the same sample.

**Influence of laser fluence**. We also studied the effect of the laser pulse fluence on the induced HCDW by increasing the laser power: Most importantly, the sharpening of the super-lattice reflection along the $l$-direction remains unaffected by changing the laser pulse fluence which means that the break-down of the $t_0$ bilayers fully takes place as soon as the switching threshold is reached. Further we found evidence that the in-plane $q$-vector of the HCDW slightly shifts towards the position of the room temperature NCCDW in-plane $q$-vector with increasing laser fluence. This trend is consistent with pump-probe electron diffraction experiments at higher temperatures[31].

## Discussion

We now turn the discussion towards the origin of the metallicity of the HCDW in the light of our observations. Our data reveal that the metallization in the HCDW coincides with a reordering of the CDW in the out-of-plane direction, as well as the forma-tion of disommensurations. Recent studies based on density functional theory provide strong evidence that the changes in the out-of-plane direction indeed play an important role for the IMT[4,23,26]. More specifically, these studies identify a key feature responsible for the insulating properties of the CCDW, namely the formation of $t_0$ bilayers characteristic for the CCDW stacking along the out-of-plane direction. Our experimental data show that these bilayers indeed collapse in response to the picosecond laser pulse as can be unambiguously seen from the vanishing double-peak structure in the intensity distribution of the super-lattice reflections along the $l$-direction (cf. Fig. 2d–i). The same breakdown of the $t_0$ bilayers also happens as a function of temperature or pressure at the CCDW–NCCDW transition, which is also accompanied by a marked change of the electrical resistivity. We therefore conclude that the rearrangement of the CDW structure in the out-of-plane direction involving the col-lapse of the $t_0$-bilayers lies at the core of the photo-induced IMT.

The formation of the $t_0$-bilayers is closely related to the hybridization between the orbitally ordered CDW layers, where each SOD represents one quasi-molecular orbital. In other words, the CCDW phase is characterized by a dimerization of SODs in the out-of-plane direction, i.e., the formation of interlayer dimer-bonds[4,23,26]. According to our observations, photoexcitation breaks exactly these interlayer dimers. Since this primarily is an electronic process involving orbital degrees of freedom, it is likely to proceed on ultra-fast electronic time scales, as observed in previous time-resolved experiments[32–35].

We would like to stress that the HCDW structure in our experiments is always long range ordered throughout the thin sample. In contrast, recent STM experiments on the HCDW show a strongly disordered network of discommensurations at the surface[17]. The sharp superlattice reflections and the absence of significant diffuse scattering in our bulk sensitive XRD measure-ments of the HCDW instead imply that the discommensurations form a highly ordered network similar to the NCCDW at room temperature. This comparison therefore implies that the photo-induced processes at the surface and in the bulk are different, which is certainly relevant with respect to the down-scaling of the sample thickness for possible technological applications.

Regarding the photo-induced IMT in the bulk, our data reveal a marked similarity between the HCDW and the NCCDW. In spite of these clear similarities, it has been argued previously that the NCCDW and HCDW are distinct states because they differ in their electrical resistivity and the frequency of the amplitude mode of the CDW[7,10]. Indeed, besides the aforementioned similarities of the HCDW and the NCCDW, the present XRD-study uncovers also an important difference between these two states, namely a different density of discommensurations. Our measurements show that the in-plane incommensurability $\delta$ of the HCDW $q$-vector is smaller than that of the room temperature NCCDW (see Table 1). This implies a smaller density of dis-commensurations in the HCDW as compared with the room temperature NCCDW[27,30]. It is likely that salient features of the discommensurations network, such as the concentration of dis-commensurations, alter the transport properties as well as the frequency of the amplitude mode. Nevertheless, it is also known that the density of discommensurations changes as a function of temperature within the NCCDW and one could speculate that the HCDW is indeed a quenched form of the NCCDW, created by the transient heating of the laser pulse and the abrupt cooling afterwards. In that spirit, HCDW and NCCDW would be two

instances of the same phase, distinguished by details of their discommensurations network.

In conclusion, we employed synchrotron XRD in combination with picosecond laser-pumping to uncover the structure of the photo-induced HCDW in 1T-TaS$_2$ nanosheets. Our measurements indicate a key mechanism of the photo-induced IMT: The collapse of interlayer dimers, which are a characteristic feature of the CCDW and which have been proposed to be the origin of its insulating properties[23,26]. Thus, the present study implies that the reorganization of the orbitally ordered layers in the out-of-plane direction, rather than in the in-plane direction lies at the core of the marked transport anomalies in 1T-TaS$_2$. On a more general level, the case at hand illustrates in a striking manner that out-of-plane correlations between the layers in Van der Waals materials can play a crucial role for their physical properties. This is not a mere complication but provides a unique way to control the electronic properties of such materials in the few layer limit.

A very intriguing and surprising feature of the studied case is that a single picosecond laser pulse can trigger a transition between two very complex and truly long-range ordered electronic states, which are the endpoints of the photo-induced IMT. How this transition proceeds on atomic length and electronic time scales still remains an open and very fascinating puzzle. A complete understanding of this unusual collective electronic phenomenon will require further time-resolved experiments.

## Methods

**Sample preparation.** The high-quality 1T-TaS$_2$ bulk single crystals used for the present study were grown from high purity elements by chemical vapor transport using iodine as transport agent. Thin films with a thickness of about 50 nm and typical lateral dimensions up to 200 μm were prepared by mechanical exfoliation. The exfoliated samples were transferred onto standard TEM silicon nitride windows with 200 nm thickness.

**XRD measurements and laser excitation.** Bulk single crystal XRD data at room temperature and 100 K were measured at a Bruker APEXII diffractometer.

XRD experiments on the exfoliated 1T-TaS$_2$ samples were performed at the beamline ID09 of the European Synchrotron Radiation Facility. The TEM silicon nitride window with the exfoliated 1T-TaS$_2$ samples were mounted on the cold finger of a specially designed continuous He-flow cryostat with Mylar windows which are transparent for both the optical laser radiation and the 18 keV x-ray radiation. The cryostat was attached to a single rotation axis. Photo-excitation of the exfoliated 1T-TaS$_2$ samples was achieved by means of a single pulse from an optical Ti-sapphire laser with a wavelength of 800 nm and a pulse duration of 1.6 ps. The laser spot size on the sample was about 400 μm (FWHM) while the x-ray beam spot size was about $60 \times 100$ μm$^2$. XRD single crystal datasets from the samples were collected using a Rayonix MX170 detector in transmission geometry. All data sets contain 120 frames with 0.5° scan width over a sample rotation of 60°. The diffraction images were transformed into reciprocal space using the CrysAlis Pro software package[36]. Reciprocal space maps as shown in Figs. 1 and 2 were produced using the Python packages numpy, matplotlib, and fabio.

## Data availability
The data that support the findings of this study are available from the corresponding author upon reasonable request.

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

## Acknowledgements

This research has been supported by the Deutsche Forschungsgemeinschaft through the SFB 1143, the Würzburg-Dresden Cluster of Excellence EXC 2147 (ct.qmat), the Graduate School GRK 1621, and Grant No. RI 2908/1-1. We also gratefully acknowledge the support provided by the DRESDEN-concept alliance of research institutions, thank T. Woike and M. Wulff for helpful discussions and the ESRF for providing the beamtime at ID09.

## Author contributions

J.G. and T.R. conceived the research project on 1T-TaS$_2$. K.H. and K.R. grew the single crystals. T.R., Q.S., M.K., F.H., G.G., N.K., and J.G. conducted the experiments. T.R., Q.S., and J.G. analyzed the results and prepared the manuscript.

## Competing interests

The authors declare no competing interests.
