## [Peer Review File · Nature Communications]

Editorial Note: Parts of this peer review file have been redacted as indicated to remove third-party material where no permission to publish could be obtained.

Reviewers' comments:

Reviewer #1 (Remarks to the Author):

The manuscript carefully describes a clear piece of work done on 1T-TaS₂. This belongs to a class of materials that show many interesting electronic properties, where charge density waves are of particular relevance here. The authors use a laser pulse to induce a transition in TaS₂, and monitor the effects of this laser pulse by the way in which it affects a charge density wave signal. This work is very elegant. I think that the work as presented merits publication in Nature Communications, and that the questions raised at the end of the paper are very relevant and important. I think that this work will be of interest to others working on CDWs, photo-induced transitions, and those interested in the lifetime of processes more generally. It is very interesting that the layer dimerization can be targeted so precisely. The manuscript is pitched to a non-specialist audience.

I have a couple of caveats. As far as I can see, in this work itself, the metallicity of the HCDW state is not measured, but only inferred from the existence of the HCDW state. While I do not really doubt that the observation here is linked to the transition to metallicity observed elsewhere, this remains an assumption in this work.

The article mentions that no stacking faults are seen. This is clearly important (especially given the arguments in the paper regarding the t_0 bilayers), but is not discussed further. This has been studied in the past, e.g. Endo et al., Solid State Comms 116, 47 (2000), and I think that this observation merits a bit more discussion in the paper. However, this should be at the authors' discretion.

To summarize, this is an important, well-written contribution, and I support publication.

Reviewer #2 (Remarks to the Author):

Stahl et al. present an interesting XRD study of CDW phases in 1T-TaS₂. The manuscript is written clearly, albeit with a few typos and a large number of abbreviations that complicate the reading for people outside the community.

The paper is mainly motivated by the quest of the origin of the CCDW-HCDW (IMT) transition. The authors first characterize the CCDW and NCCDW phases by XRD and emphasize the dimerization observed by XRD. They then show the XRD data of the thermal NCCDW-CCDW and the photoinduced CCDW-HCDW transition and find that the dimerization is lost in the HCDW phase, which can be reestablished (data not shown) upon heating to 40 K. The authors argue that their results imply that the dimerization is responsible for the insulating properties of the CCDW phase and that "out-of-plane correlations" can significantly modify physical properties in the whole material class.

In my opinion, the manuscript may be publishable in the present form, but NatCommun does not seem to be the right journal, as the main result is quite specific. I can, however, imagine that the manuscript could be rewritten such that the broader implications of the results become clearer. This would require major revision of the manuscript.

Whatever the authors choose (even if they send the manuscript elsewhere), I strongly recommend considering the following points when revising the manuscript:

1. Beyond using the established recipe for HCDW formation, the authors do not seem to give any evidence for the actual formation of this phase. What are the characteristics of this phase? How could

they demonstrate that they actually reach it? This is particularly an issue, because the NCCDW and the HCDW are so very similar in terms of their XRD features. I recommend elaborating on this more.

2. Previous studies suggest that the dimerization upon cooling leads to the MIT in the thermal (NCCDW-CCDW) transition. In their work, the authors now show that this dimerization is lost in the HCDW phase. Based on this, the authors claim that they “disclose a key mechanism in the photo-induced IMT”. However, in order to have a mechanism a cause-and-consequence proof needs to be presented. The IMT (CCDW-HCDW) is accompanied by the loss of dimerization, yes. But is it the cause? The current discussion of this is not convincing.

3. Similarly, the authors claim that their work implies that the out-of-plane direction “lies at the core” of the transport anomalies. There is a change in this direction, yes. But which observation shows that this is the mechanism?

4. The authors speak about correlations between the layers and collective phenomena, very fashionable terms. But what are the observations that support a collective, correlated & cooperative mechanism across the layers? What would be the interaction that leads to a collective phenomenon? The layers are only bound by vdW interaction. Isn't the presence/absence of a certain periodicity likely the origin of the different transport behaviours? Why involve correlations?

5. In the abstract, the authors promise to determine “the nature of the so-called hidden quantum state”. They do characterize some of its properties, yes. But in how far do they unveil its nature (see the points above)?

6. I would appreciate a more in-depth discussion of the authors' view of the photoinduced transition. Why can't the laser pulse lead to an abrupt heating of the sample, which is sufficient to drive the system into the HCDW, which seems like an incomplete NCCDW? Could it be that the much faster cooling after photoexcitation compared to the slow cooling from 300 to 4 K somehow freezes an incomplete NCCDW? This could be metastable and would be consistent with the low 40 K transition temperature for the HCDW-CCDW transition.

In summary, the present version of the manuscript does hold what the title promises: The demonstration of a photo-induced loss of dimerization. –Whether the photoinduced state is really the HCDW phase of the material remains somehow vague. However, the authors *claim* at many instances in text that they show that this loss is the source of the electronic transition, but do not *demonstrate* that the dimerization loss is the cause of the latter. It simply occurs concurrently. Therefore, in the present form I cannot recommend this article for publication in NatCommun. From my point of view, the authors need to either substantiate their claims or reformulate. Nevertheless, I would like to stress that their observations are really interesting and, in my opinion, could attract a wide readership if “more meat” is added to the discussion.

General comments:

- fewer abbreviations would facilitate reading

- The use of CDW, HCDW etc. is not consistent. Sometimes the authors seem to mean “charge density wave” (as introduced), sometimes CDW-order, sometimes CDW-stacking etc

- In some instances, the authors speak about CDW-order perpendicular to the layers of the material. Although this kind of terminology might have spread across the community, I doubt that this is an

entirely correct way of discussing the 2D-CDWs in a vdW-layered material, while the use of commensurate vs incommensurate seems more appropriate until it is shown that the CD perpendicular to the planes is altered in the insulating phase and thereby establishing the different electronic properties. If only unaltered 2D-CDWs are stacked differently and thereby generate a new periodicity, I believe, the Peirls picture is not the correct one. – But I am willing to be convinced otherwise.

- It seems obvious that the paper was originally sent to an APS journal (likely PRL). The authors should invest some more effort to structure the manuscript according to the present journal style. It seems like they have simply added the “results” and “discussion” section titles, but have not adapted the preceding and subsequent paragraphs to be the last and first paragraphs of a section. It is for example somewhat odd that the 1st sentence of the results section directly links to the last of the general part.

Here are some minor things I came across:

p1, 2nd paragraph, first sentence: 2x particularly

p1, 2nd column, 2nd sentence: in-particular should read in particular

p3, 1st paragraph: In fact, CDW-stacking...

p4, 2nd column: should read: ...clearly shows...truly long-range ordered...

p6, 2nd column: nm and um non-italic

Response to the referees:

Referee #1:

We thank referee #1 for evaluating our work and appreciate the referees' very positive assessment:

“The manuscript carefully describes a clear piece of work done on 1T-TaS₂. This belongs to a class of materials that show many interesting electronic properties (...) charge density waves are of particular relevance here. The This work is very elegant. I think that the work as presented merits publication in Nature Communications, and that the questions raised at the end of the paper are very relevant and important. I think that this work will be of interest to others working on CDWs, photo-induced transitions, and those interested in the lifetime of processes more generally. It is very interesting that the layer dimerization can be targeted so precisely. The manuscript is pitched to a non-specialist audience.”

He or she recommends our manuscript for publication in the present form. Additionally, referee #1 lists two comments:

“As far as I can see, in this work itself, the metallicity of the HCDW state is not measured, but only inferred from the existence of the HCDW state. While I do not really doubt that the observation here is linked to the transition to metallicity observed elsewhere, this remains an assumption in this work.”

This important point is very similar to the main concern expressed by the second referee. Accordingly, we refer to the corresponding section in the reply to the second referee.

Further referee #1 notes:

“The article mentions that no stacking faults are seen. This is clearly important (especially given the arguments in the paper regarding the t₀ bilayers), but is not discussed further. This has been studied in the past, e.g. Endo et al., Solid State Comms 116, 47 (2000), and I think that this observation merits a bit more discussion in the paper. However, this should be at the authors' discretion.”

We agree with the referee that this discussion came a bit short in the previous version of the manuscript. With the statement *“It should be noted that merely the superlattice reflections show diffuse scattering stripes while the Bragg-peaks remain sharp indicating*

Figure S1: Peak profile of a typical Bragg peaks measured on a bulk single crystal with a lab single crystal diffractometer (a) and on the exfoliated thin flakes measured at the synchrotron (b). Peak profiles in the out-of-plane l -direction (blue) and the in-plane k -direction (green) are shown

the absence of more conventional stacking faults in the average structure.” we wanted to emphasize that the partially disordered CDW stacking should not be confused with crystallographic stacking faults in the average 1T structure like embedded 2H layers. The peak width of the Bragg peaks is resolution limited and basically the same for the out-of-plane l -direction and the in-plane k -direction indicating a lower bound of the domain size of about $2\pi/0.04 \text{ \AA}^{-1} \approx 160 \text{ \AA}$ (roughly 25 layers), as illustrated in Fig. S1 (a). This should certainly just be seen as a lower limit given by the low q -space resolution of the laboratory single crystal diffractometer. For comparison, Fig. S1 (b) illustrates the much better q -resolution of our synchrotron study (shown are the peak profiles of a Bragg peak in the h and l direction measured on an exfoliated flake). In the h -direction the FWHM is about 0.01 \AA^{-1} while in the l direction it is again about 0.04 \AA^{-1} resulting in a domain size of about 25 layers which is indeed comparable with the thickness of these exfoliated flakes.

To make this point more clear we added a bar to Fig. 1 and Fig. 2. of the manuscript to indicate the FWHM of the Bragg peak in the l -direction.

Referee #2:

We thank referee #2 for the critical reading of our manuscript and the constructive comments which helped to improve our manuscript. We are very happy to hear that he or she finds our results interesting and of potential interest for a wide readership.

“Stahl et al. present an interesting XRD study of CDW phases in 1T-TaS₂. (...) In my opinion, the manuscript may be publishable in the present form (...). I would like to stress that their observations are really interesting and, in my opinion, could attract a wide readership if “more meat” is added to the discussion.”

However, referee #2 does not recommend the original version of our article for publication in Nature Communications.

“(...) but NatCommun does not seem to be the right journal, as the main result is quite specific. I can, however, imagine that the manuscript could be rewritten such that the broader implications of the results become clearer. This would require major revision of the manuscript.”

At this point we have to disagree with referee #2. We are convinced that Nature Communications is indeed the right place for our study as the photo-induced hidden state of 1T-TaS₂ is a recurring topic in this journal with at least five papers published about this problem over the last few years [1–5]. We believe that our work will be appreciated by the readership of Nature Communications as it adds important new insights about the hidden state.

The main point of technical criticism raised by referee #2 is formulated in his or her **1st comment** and concerns the lack of verification that the photo-induced state in our experiment is indeed the same state as the one that is known as the hidden state of 1T-TaS₂.

“Beyond using the established recipe for HCDW formation, the authors do not seem to give any evidence for the actual formation of this phase. What are the characteristics of this phase? How could they demonstrate that they actually reach it? This is particularly an issue, because the NCCDW and the HCDW are so very similar in terms of their XRD features. I recommend elaborating on this more.”

This point was also raised, in a similar way, by the first referee. We thank the referees for bringing up this important point and we agree that a measurement of the resistivity would be the most direct way to prove the metalization of the photo-induced phase. However, we would like to stress that these synchrotron experiments are highly non-standard and very challenging. As far as we know, our study indeed demonstrates, for the first time, the use of x-rays to measure subtle photo-induced changes of a faint superstructure in thin exfoliated flakes of a layered material. We believe that the demon-

stration of this experimental technique alone would merit publication in a high-impact journal like Nature Communications. This was also acknowledged by the first referee:

“(...) It is very interesting that the layer dimerization can be targeted so precisely. (...)”

Due to the complexity and very challenging nature of the presented experiments, we can, unfortunately, not provide in-situ resistivity measurements. However, in what follows we will provide extensive argumentation which conclusively shows that the photo-induced state in our experiments is indeed the so-called hidden state of 1T-TaS₂:

1. The most important characteristic of the HCDW is perhaps that it relaxes back to the CCDW phase already at relatively low temperatures of about 40 - 70K which is significantly below the thermal NCCDW-CCDW transition at around 200 K (see for instance Fig. 1. in [6] L. Stojchevska et al., “Ultrafast switching to a stable hidden quantum state in an electronic crystal”, Science **344**, 177–180 (2014), which we reproduce for convinience in Fig. S2).

[Redacted]

Figure S2: Resistivity switching of 1T-TaS₂ by a 35-fs laser pulse at 800 nm. Taken from [6] L. Stojchevska et al., “Ultrafast switching to a stable hidden quantum state in an electronic crystal”, Science **344**, 177–180 (2014).

Referee #2 is correct in criticizing that we did not show the related data in the previous version of our manuscript. Therefore, we present the data for two successive laser-pulse/heating cycles on the same sample in Fig. S3. As can be seen in

Fig. S3, we observe that our samples relax back to the CCDW phase in precisely the same temperature range at about 40 K. Accordingly, we think it is valid to conclude that the photo-induced state in our experiment is the same as the photo-induced HCDW that has been found in previous transport measurements. In the data shown in Fig. S3 it can be seen that the switching is not always complete. For that specific sample we observed a partial switching indicated by the presence of diffraction features belonging to both the HCDW and CCDW (see Fig. S3 (c-e)). For the second laser-pulse/heating cycle (Fig. S3 (g-l)), where we used a higher laser fluence, the switching was essentially complete and the CCDW peaks reappear at slightly higher temperatures than in the first laser-pulse/heating cycle.

In the original version of the manuscript we omitted the data of the thermally driven HCDW-CCDW transition in order to keep the article focused. However, we agree with the referee that this is an important observation and decided to add a compact version of Fig. S3 as a third figure, which demonstrates the HCDW-CCDW transition at around 40 K, to the manuscript.

2. The photo-induced state in our experiments is persistent. This is also a defining property of the HCDW. Indeed, only the non-volatility of the HCDW makes it possible to measure its full 3D structure as done in our study. The persistence of the photo-induced HCDW in our experiments can also be seen in Fig. S3 (c) and (d) which show the data for two consecutive measurements one hour apart.
3. As noted by both referees we followed the established recipe to create the hidden state. It is, hence, rather unlikely that we created a completely different state that, nevertheless, has property 1) and 2) in common with the HCDW.

Given the above arguments we believe that our conclusion, namely that the photo-induced state probed by our X-ray diffraction experiments is identical to the HCDW, is well founded. However, we agree that it is not proven directly by a measurement of the electrical resistivity. Therefore, we revised the corresponding paragraphs in the manuscript to emphasize that our well-justified assumption that we probe the HCDW is based on the marked characteristics as explained above, rather than a direct in-situ resistivity measurement.

Apart from this main point of criticisms referee #2 lists five additional comments which we will address one by one in the following paragraphs:

2nd comment: The second comment criticizes that we show a correlation, rather than a causality between the loss of dimerization and the photo-induced metallization:

Figure S3: Evolution of the superstructure peak profiles in the l -direction for two full laser-pulse/heating cycles. (a): The NCCDW superstructure peak at room temperature. (b): The diffuse CCDW double peak at 4 K. (c): Peak profile after photo-excitation at 4 K. Coexistence of CCDW peak and HCDW peak indicating only partial switching. (d): Same as (c) after waiting for 1 hour demonstrating the persistence of the HCDW. (e): Peak profile at 20 K. (f): Peak profile at 40 K where the HCDW peak vanishes. (g): Peak profile after cooling to 4 K again. (h) Peak profile after second laser pulse. This time the switching was more complete. (i-l) Second warming cycle. The CCDW peaks reappear at around 40 K in (j).

“Previous studies suggest that the dimerization upon cooling leads to the MIT in the thermal (NCCDW-CCDW) transition. In their work, the authors now show that this dimerization is lost in the HCDW phase. Based on this, the authors claim that they “disclose a key mechanism in the photo-induced IMT”. However, in order to have a mechanism a cause-and-consequence proof needs to be presented. The IMT (CCDW-HCDW) is accompanied by the loss of dimerization, yes. But is it the cause? The current discussion of this is not convincing.”

We think that it is very hard to strictly prove a causality like “metallization follows from the collapse of dimers” purely experimentally. The standard scientific approach to this issue is to construct hypothesis based on theoretical models and to verify or falsify them experimentally. This is exactly what we did. In our opinion, the theoretical prediction that a dimerized phase is insulating while the non-dimerized phase is a metal in conjunction with our experimental finding that the dimerization breaks down upon photo-excitation provides strong evidence and strong experimental support of the notion that this dimerization is indeed the key mechanism. This is an important contribution to the ongoing discussion, which we think will be appreciated by the broad readership of Nature Communications.

3rd comment: The third comment is a continuation of the first comment.

“Similarity, the authors claim that their work implies that the out-of-plane direction “lies at the core” of the transport anomalies. There is a change in this direction, yes. But which observation shows that this is the mechanism?”

As explained above, we argue that comparison to theoretical predictions is a methodologically sound means of identifying key mechanisms. In the present case, a theoretical prediction exists which relates the changes along the out-of-plane directions to changes of the transport properties. The experimental observation that these changes indeed occur in the HCDW consequently suggest that changes of the CDW-layer stacking drive the insulator-to-metal transition. In our manuscript we clearly described how we reached this conclusion: *“More specifically, these studies identify a key feature responsible for the insulating properties of the CCDW, namely the formation of t_0 bilayers characteristic for the CCDW stacking along the out-of-plane direction. Our experimental data show that these bilayers indeed collapse in response to the picosecond laser pulse (...) We therefore conclude that the rearrangement of the CDW structure in the out-of-plane direction involving the collapse of the t_0 -bilayers lies at the core of the photo-induced IMT.”*

Another strong argument in favor of interlayer effects as the driving mechanism is the isostructural compound 1T-TaSe₂. This material develops the exact same in-plane $\sqrt{13} \times \sqrt{13}$ CCDW superstructure at low temperatures. However, as opposed to 1T-TaS₂ the CDW stacking is found to be of the type t_{11} with long-range order and the transport properties of 1T-TaSe₂ in the CCDW are metallic [7]. This clearly shows that correlations of the CDW in the out-of-plane direction indeed play a major role for the transport properties of the materials.

4th comment: The fourth comment of referee #2 refers to the terminology we use to describe the different CDW states:

“The authors speak about correlations between the layers and collective phenomena, very fashionable terms. But what are the observations that support a collective, correlated & cooperative mechanism across the layers? What would be the interaction that leads to a collective phenomenon? The layers are only bound by vdW interaction. Isn’t the presence/absence of a certain periodicity likely the origin of the different transport behaviours? Why involve correlations?”

At this point we think it is important to note that the present case is very unusual in the sense that a single laser pulse transforms one long-range ordered state into a different long-range ordered state. Usually, photo excitation leads to a melting of electronic order and a recovery of the original state on some characteristic time-scales. In order to form a long-range ordered state, collective and cooperative mechanism are needed.

We agree that the problem can be reduced to the occurrence of different periodicities along the c -direction causing different transport behaviours. However, that does not explain the mechanism that leads to these different periodicities. Only interactions or correlations between the CDWs in neighboring layers can stabilize these different periodicities along the c -direction. In order to emphasize this point, namely that there must be significant interaction between layers, we used the term correlation.

In fact, van der Waals interaction itself is a prototypical electron correlation effect as it arises from long-range correlations between fluctuating electron-hole pairs which is also the reason why van der Waals bond lengths are usually not well described by standard density functional theory. In that sense we do not see any problem in using the terms correlation, collective and cooperative in the present context.

5th comment: The fourth comment concerns a sentence from the abstract of our manuscript:

“In the abstract, the authors promise to determine “the nature of the so-called hidden quantum state”. They do characterize some of its properties, yes. But in how far do they unveil its nature (see the points above)?”

To our knowledge our work is the first that reveals the microscopic structure and, hence, determines key characteristics of the HCDW which, in our opinion, could well be called “unveiling its nature”. To emphasize this point our study unambiguously shows that:

1. the HCDW is a long-range ordered CDW phase,
2. the interlayer dimerization found in the CCDW is lost in the HCDW,
3. the HCDW is very similar to the NCCDW.

Nevertheless, based on the referee’s comment we rephrased that sentence in the abstract to “(...) Here we characterize the hidden quantum state of 1T-TaS₂ by means (...)”

6th comment: Finally referee #2 remarks:

“I would appreciate a more in-depth discussion of the authors’ view of the photoinduced transition. Why can’t the laser pulse lead to an abrupt heating of the sample, which is sufficient to drive the system into the HCDW, which seems like an incomplete NCCDW? Could it be that the much faster cooling after photoexcitation compared to the slow cooling from 300 to 4 K somehow freezes an incomplete NCCDW? This could be metastable and would be consistent with the low 40 K transition temperature for the HCDW-CCDW transition.”

We like to thank the referee for bringing up this important point. As described in our manuscript we agree with the referee’s point of view that the HCDW is very similar to the NCCDW and that both could indeed be two instances of the same phase. In our manuscript we write: “(...) Regarding the photo-induced IMT in the bulk, our data reveal a marked similarity between the HCDW and the NCCDW. In spite of these clear similarities, it has been argued previously that the NCCDW and HCDW are distinct states because they differ in their electrical resistivity and the frequency of the amplitude mode of the CDW. Indeed, besides the aforementioned similarities of the HCDW and the NCCDW, the present XRD-study uncovers also an important difference between these two states, namely a different density of discommensurations. Our measurements show that the in-plane incommensurability δ of the HCDW q -vector is smaller than that of the room temperature NCCDW (see Table I), implying a smaller density of discommensurations in the HCDW as compared to the room temperature NCCDW. It is likely

that salient features of the discommensurations network, such as the concentration of discommensurations, alter the transport properties as well as the frequency of the amplitude mode. ”

However, since we cannot derive a full structure refinement of the HCDW from our data, we formulated this argument in a speculative way. In the revised manuscript we added a paragraph to emphasize the possibility of the HCDW being a quenched NCCDW.

Referee #2 summarizes his or her point of view with the following comment:

*“In summary, the present version of the manuscript does hold what the title promises: The demonstration of a photo-induced loss of dimerization. –Whether the photoinduced state is really the HCDW phase of the material remains somehow vague. However, the authors *claim* at many instances in text that they show that this loss is the source of the electronic transition, but do not *demonstrate* that the dimerization loss is the cause of the latter. It simply occurs concurrently. Therefore, in the present form I cannot recommend this article for publication in NatCommun. From my point of view, the authors need to either substantiate their claims or reformulate. Nevertheless, I would like to stress that their observations are really interesting and, in my opinion, could attract a wide readership if “more meat” is added to the discussion.”*

As explained in detail above, our line of arguments follows a commonly used strategy which is based on the verification of theoretical predictions. As suggested by the referee #2, we expanded the corresponding discussion in the paper and also reformulated our claims. We hope that he or she will be satisfied that this does indeed add “more meat” to the discussion in the revised version of our manuscript.

The referee closes with some general comments:

“fewer abbreviations would facilitate reading (...) The use of CDW, HCDW etc. is not consistent. Sometimes the authors seem to mean “charge density wave” (as introduced), sometimes CDW-order, sometimes CDW-stacking etc”

We tried to reduce the number of abbreviations and also checked the consistency of the used terminology.

“In some instances, the authors speak about CDW-order perpendicular to the layers of the material. Although this kind of terminology might have spread

across the community, I doubt that this is an entirely correct way of discussing the 2D-CDWs in a vdW-layered material, while the use of commensurate vs incommensurate seems more appropriate until it is shown that the CD perpendicular to the planes is altered in the insulating phase and thereby establishing the different electronic properties. If only unaltered 2D-CDWs are stacked differently and thereby generate a new periodicity, I believe, the Peirls picture is not the correct one. – But I am willing to be convinced otherwise.”

With “CDW-order perpendicular to the layers” we mean the stacking arrangement of the CDW-order along the direction perpendicular to the layers, i.e. the CDW-stacking. We agree that the simple Peirls picture is not correct. We rather think that interlayer hybridization of the orbital order that accompanies the in-plane CDW is a key mechanism for the stabilization of the different stacking arrangements.

“It seems obvious that the paper was originally sent to an APS journal (likely PRL). The authors should invest some more effort to structure the manuscript according to the present journal style. It seems like they have simply added the “results” and “discussion” section titles, but have not adapted the preceding and subsequent paragraphs to be the last and first paragraphs of a section. It is for example somewhat odd that the 1st sentence of the results section directly links to the last of the general part.”

Although we used the REVTeX package for preparing the manuscript it has not been submitted elsewhere. Following the referee’s suggestions we have reorganized the last and first paragraphs of the introduction section and results section, respectively. We believe that our study is highly suited indeed for publication in Nature Communications. The photo-induced hidden state of 1T-TaS₂ is a well-established and recurring theme in this journal which has featured several high profile papers on this topic over last few years [1–5]. Therefore, we anticipate that the readership of Nature Communications will very much appreciate the new level of microscopic insight on the hidden state in this topical material that is provided by our x-ray study.

Summary of changes:

- Change 1: We added a third figure (Fig. 3) which clearly shows the characteristic temperature evolution of the HCDW for two successive laser pulse/heating cycles. This is a major addition based on the second referees' suggestion. Along with that we largely expanded the discussion on why it is valid to assume that the photo-induced state in our study is the same as the hidden state previously characterized by transport measurements.
- Change 2: We changed a sentence in the abstract from "Here we determine the latter ..." to "Here we characterize the hidden ..."
- Change 3: We added indicators to Fig. 1 and Fig. 2 showing the full width at half maximum from typical Bragg peaks.
- Change 4: We rearranged the last paragraph of the introduction section and the first paragraph of the results section to enhance the readability.
- Change 5: We added a paragraph concerning the similarities of the HCDW and the NCCDW and further discuss the possibility of a quenched NC-CDW.
- Change 6: We added a citation for a recently appeared study which discusses the role of the discommensurations network for the metallization of the NCCDW.
- Change 7: We corrected a number of typos.
- Change 8: We checked the consistence of the used terms CDW-order vs. CDW-stacking.
- Change 9: We reformulated our claims concerning the relation between loss of dimerization and metallization.
- Change 10: We added a data availability statement.

References

- [1] I. Vaskivskiy, I. A. Mihailovic, S. Brazovskii, J. Gospodaric, T. Mertelj, D. Svetin, P. Sutar, and D. Mihailovic, "Fast electronic resistance switching involving hidden charge density wave states", *Nat Commun* **7**, 11442 (2016).
- [2] L. Ma, C. Ye, Y. Yu, X. F. Lu, X. Niu, S. Kim, D. Feng, D. Tomanek, Y.-W. Son, X. H. Chen, and Y. Zhang, "A metallic mosaic phase and the origin of Mott-insulating state in 1T-TaS₂", *Nat Commun* **7**, 10956 (2016).

- [3] D. Cho, S. Cheon, K.-S. Kim, S.-H. Lee, Y.-H. Cho, S.-W. Cheong, and H. W. Yeom, “Nanoscale manipulation of the Mott insulating state coupled to charge order in 1T-TaS₂”, Nat Commun **7**, 10453 (2016).
- [4] D. Cho, G. Gye, J. Lee, S.-H. Lee, L. Wang, S.-W. Cheong, and H. W. Yeom, “Correlated electronic states at domain walls of a mott-charge-density-wave insulator 1T-TaS₂”, Nat Commun **8** (2017).
- [5] J. W. Park, G. Y. Cho, J. Lee, and H. W. Yeom, “Emergent honeycomb network of topological excitations in correlated charge density wave”, Nat Commun **10** (2019).
- [6] L. Stojchevska, I. Vaskivskyi, T. Mertelj, P. Kusar, D. Svetin, S. Brazovskii, and D. Mihailovic, “Ultrafast switching to a stable hidden quantum state in an electronic crystal”, Science **344**, 177–180 (2014).
- [7] G. Wieggers, J. de Boer, A. Meetsma, and S. van Smaalen, “Domain structure and refinement of the triclinic superstructure of 1T-TaSe₂ by single crystal x-ray diffraction”, Zeitschrift für Kristallographie/International journal for structural, physical, and chemical aspects of crystalline materials **216**, 45–50 (2001).

REVIEWERS' COMMENTS:

Reviewer #1 (Remarks to the Author):

In my previous report, I raised two concerns. The first was concerned with the fact that the metallicity of the HCDW state is assumed, rather than directly measured. The authors have addressed this concern carefully in their rebuttal, accepting the main statement. The paper has been modified, to strengthen the authors' argument that this assumption is valid, as well as laying out more carefully the fact that this is an assumption. I also appreciated the additions to the manuscript in response to the other reviewer's 6th comment, regarding the possible effects of laser heating.

My second concern was with the discussion of stacking faults. The authors provide clean evidence in their rebuttal to show that the synchrotron experiment gives clean Bragg peaks in the (010) direction. I note that the scan along the (001) direction shows an apparent double peak structure. However, I accept the broad thrust of the authors' argument.

2 Second revision round

2.1 Reviewers comments to the Authors

2.1.1 Reviewer #1

In my previous report, I raised two concerns. The first was concerned with the fact that the metallicity of the HCDW state is assumed, rather than directly measured. The authors have addressed this concern carefully in their rebuttal, accepting the main statement. The paper has been modified, to strengthen the authors' argument that this assumption is valid, as well as laying out more carefully the fact that this is an assumption. I also appreciated the additions to the manuscript in response to the other reviewer's 6th comment, regarding the possible effects of laser heating.

My second concern was with the discussion of stacking faults. The authors provide clean evidence in their rebuttal to show that the synchrotron experiment gives clean Bragg peaks in the (010) direction. I note that the scan along the (001) direction shows an apparent double peak structure. However, I accept the broad thrust of the authors' argument.

2.2 Response to the Referee's comments

2.2.1 Reviewer #1

We thank the Referee for carefully reading our revised manuscript. We fully agree with him or her that the new version points out more clearly that the metallicity of the photo-induced state in our study is an assumption that is, however, well-founded. We are happy to hear that the referee appreciates our addition to the manuscript concerning the possible effects of laser heating in response to the 6th comment of referee #2.

We are also happy to hear that the referee is satisfied with the data we provided in order to show the high crystalline quality of our thin flake samples. Concerning the synchrotron experiment peak profile of the Bragg peak in the (001) direction (c.f. Figure S1 (b)) we would like to note that the peak profiles are extracted from 2D detector images which makes a quantitative analysis of slight features in the peak profile like the subtle shoulder in Figure S1 (b) difficult.